# Human Immunodeficiency Virus and Clonal Hematopoiesis

**DOI:** 10.3390/cells12050686

**Published:** 2023-02-22

**Authors:** Stamatia C. Vorri, Ilias Christodoulou, Styliani Karanika, Theodoros Karantanos

**Affiliations:** 1Division of Pediatric Oncology, Department of Oncology, Johns Hopkins University Hospital, Baltimore, MD 21287, USA; 2Department of Internal Medicine, University of Pittsburgh, Pittsburgh, PA 15260, USA; 3Division of Infectious Diseases, Department of Medicine, Johns Hopkins University, Baltimore, MD 21287, USA; 4Division of Hematologic Malignancies, Department of Oncology, Johns Hopkins University Hospital, Baltimore, MD 21287, USA

**Keywords:** HIV, human immunodeficiency virus, clonal hematopoiesis, non-AIDS related comorbidities

## Abstract

The evolution of antiretroviral therapies (ART) has tremendously improved the life expectancy of people living with human immunodeficiency virus (HIV) (PLWH), which is currently similar to the general population. However, as PLWH are now living longer, they exhibit various comorbidities such as a higher risk of cardiovascular disease (CVD) and non-acquired immunodeficiency syndrome (AIDS)-defined malignancies. Clonal hematopoiesis (CH) is the acquisition of somatic mutations by the hematopoietic stem cells, rendering them survival and growth benefit, thus leading to their clonal dominance in the bone marrow. Recent epidemiologic studies have highlighted that PLWH have a higher prevalence of CH, which in turn is associated with increased CVD risk. Thus, a link between HIV infection and a higher risk for CVD might be explained through the induction of inflammatory signaling in the monocytes carrying CH mutations. Among the PLWH, CH is associated with an overall poorer control of HIV infection; an association that requires further mechanistic evaluation. Finally, CH is linked to an increased risk of progression to myeloid neoplasms including myelodysplastic syndrome (MDS) and acute myeloid leukemia (AML), which are associated with particularly poor outcomes among patients with HIV infection. These bidirectional associations require further molecular-level understanding, highlighting the need for more preclinical and prospective clinical studies. This review summarizes the current literature on the association between CH and HIV infection.

## 1. Introduction

Although the life expectancy of people living with HIV (PLWH) has improved steeply in the past decades with the introduction of safe and effective antiretroviral therapy (ART), HIV infection continues to be associated with various comorbidities [1,2], including but not limited to cardiovascular disease (CVD), accelerated aging [3,4] and non-acquired immunodeficiency syndrome (AIDS)-related neoplasms [5]. Given that ART suppresses the HIV viral load to an undetectable level, the majority of these comorbidities have been largely attributed to chronic low-level inflammation [6,7], altered T-cell biology [8], ART adverse effects [9,10] and the prothrombotic phenotype of PLWH [10,11]. The current research focuses on early risk assessment, identifying very high-risk individuals, and preventing these comorbidities. Preclinical and clinical research aims to understand the mechanisms mediating the association of HIV infection with these comorbidities.

Clonal hematopoiesis (CH) is the process of acquiring somatic mutations in the genes that affect the function of the hematopoietic stem cells (HSC) by providing a growth advantage to the mutated cells over the unmutated counterparts, resulting in the dominance of the malignant clone in the bone marrow [12]. CH without evidence of hematologic malignancy, dysplasia, or cytopenia with a variant allele frequency (VAF) of at least 2% is defined as clonal hematopoiesis of indeterminate potential (CHIP) [12,13]. In contrast, the CH associated with low counts in at least one cell lineage is defined as clonal cytopenia of unknown significance (CCUS) [13]. The prevalence of CH increases with aging and is estimated to be 15% among individuals older than 70 years [14]. CH has been associated with worse overall outcomes, such as a higher incidence of atherosclerosis and CVD, which is mediated by the induction of innate immune signaling and chronic low-level inflammation [15] and the increased risk of progression to myeloid neoplasms such as myelodysplastic syndrome (MDS) or acute myeloid leukemia (AML) [16,17]. The most common CH-associated mutations are in the *DNMT3A* and *TET2* genes, which encode the epigenetic regulators and cause altered transcriptional profiles in differentiated monocytes and macrophages, leading to the upregulation of the inflammatory cytokines such as IL-1β [18]. Less common mutations associated with CH are mutations in the genes such as *ASXL1* and *TP53*, which provide more prominent survival and growth advantage to the mutated hematopoietic cells, introducing a higher risk of progression to myeloid neoplasms [12,14]. 

Patients with HIV previously presented with severe cytopenias in the setting of uncontrolled viremia and chronic infections, but the introduction of efficacious ART has dramatically decreased the incidence of these abnormalities. However, mild cytopenias, particularly anemias, continue to be commonly described in PLWH [19,20,21]. Recent studies highlight that CH is more common among PLWH compared to the general population [18,22,23] and is associated with altered viral control [24] and increased risk of CVD [25], providing a novel link between HIV infection and chronic inflammation in PLWH via CH. Lastly, it has been demonstrated that HIV infection alters the outcomes of patients with advanced myeloid stem cell neoplasms [26]. Thus, a bidirectional effect between HIV infection and abnormal hematopoiesis exists. 

This review summarizes the current findings on the prevalence of CH among PLWH and the implication of CH in the control of HIV. 

## 2. Clonal Hematopoiesis in Patients Living with HIV

Dharan et al. analyzed the prevalence and the characteristics of CH in 220 PLWH and 226 HIV-negative Australian adults over the age of 55 enrolled in the ARCHIVE study by performing targeted sequencing of genomic DNA extracted from the peripheral blood [18]. PLWH had a significantly higher prevalence of CH than the non-HIV controls (28.2% vs. 16.8%), with most of the mutations observed in the *DNMT3A*, *TET2*, and *ASXL1* genes. The difference in the prevalence of the higher-risk *ASXL1* mutations between the PLWH and the non-HIV controls was more prominent [18]. It was noted that the sub-analysis for VAF, gender and sexual orientation, ancestry, BMI, the extent of smoking exposure and alcohol use, recreational and injection drug use, annual household income, type of health insurance coverage, and HIV-specific characteristics showed no significant correlation [18]. These findings suggest that PLWH have a higher prevalence of CH and its associated mutations than non-HIV infected individuals. These observational conclusions are limited by the relatively small number of patients studied. Interestingly, the authors showed no correlation between the duration of HIV and the VAF or the presence of more than one mutation [18]. In this study population, the prevalence of cardiovascular comorbidities was similar between the PLWH and the controls (64.1 vs. 65.5%). The authors evaluated the prevalence of a mutation in the IL-6 receptor (*IL6R* p.Asp358Ala), suggested by a different study to play a role in CH-related cardiovascular risk, but found no difference between the two groups [18]. No other correlation or sub-analysis of CVD was made in this study. It is important to note that while the ARCHIVE study was designed and powered as a prospective cohort study with a 10-year follow-up of participants, these results were published in a single-time-point cross-sectional manner [18]. 

In another study, Bick et al. assessed the prevalence of CH in a multi-ethnic, randomly selected sample of 600 PLWH enrolled in the Swiss HIV Cohort Study (SHCS), aged between 21 and 83, and 8111 individuals with available exome sequences enrolled in the Atherosclerotic risk in the Community study (ARIC), aged between 45 and 84, as the population controls [22]. A significant association between HIV case status and CHIP was observed [22]. To account for the demographic imbalance of the cases and controls, the authors performed a 1:5 propensity matching strategy to select a subgroup of 230 cases and 1002 controls with similar baseline characteristics. This subset analysis detected CHIP in 7% of the exomes from the PLWH but only in 3% of the controls [22]. It is noted that the authors demonstrated a positive correlation between the prevalence of CH and ART duration but not with HIV infection duration [22]. It is possible that this correlation still underlies the importance of the chronicity of low-level inflammation in PLWH, even under ART, as a mechanism of chronic stress in the hematopoietic stem cells. This correlation may reflect that the most critical risk factor for the development of CH is advanced age. 

Van der Heijden et al. performed a cross-sectional cohort study comparing the prevalence of CH in 217 individuals; primarily men, PLWH on stable ART from the 200 HIV cohort (between 24 and 74 years) with the prevalence of CH in a cohort of overweight individuals and a cohort of age- and sex-matched population controls [24]. The authors confirmed that the probability of CHIP was significantly higher in the PLWH compared to the HIV-uninfected overweight controls [24]. Regarding mutations in specific genes, the proportion of the CH mutations in genes other than *DNMT3A*, and specifically *JAK2*, *STAT3*, and *TP53*, was larger in the PLWH [24]. Furthermore, the authors performed a mutational signature analysis showing that the clock-like and the reactive oxygen species signatures contributed uniquely to CH in the PLWH. In contrast, DNA mismatch repair signatures contributed uniquely to CH in the HIV-uninfected controls [24]. C>A mutations were identified as contributing to CH in the PLWH with prior exposure to zidovudine (AZT) but not in the unexposed individuals [24]. These results highlighted that a different mutational process potentially drives CH in PLWH and that the underlying biology of CH pathogenesis is different between PLWH and non-infected individuals. Finally, this study highlighted that within the PLWH group, CH mutations carriers had elevated coagulation markers (D-dimer and von Willebrand Factor) compared to the PLWH without the CH mutation [24], which further suggests that CH in PLWH is associated with an increased risk of thrombosis and potentially worse cardiovascular outcomes.

In a more recent cross-sectional study, Wang et al. examined the differences in CHIP and coronary artery disease prevalence between PLWH and non-HIV-infected individuals [23]. The authors performed next-generation sequencing in genomic DNA extracted from the peripheral blood mononuclear cells of 118 men (86 PLWH and 32 non-HIV-infected individuals) aged between 42 and 70 from the Baltimore-Washington D.C. center of the Multicenter AIDS Cohort Study (MACS) cohort who had coronary computed tomography angiography (CTA) and a measurement of multiple serologic inflammatory biomarkers [23]. Using a VAF cut-off of 1% and excluding germline variants, the PLWH were almost two-fold more likely to have CH than the non-infected controls (64% vs. 38%) [23]. Using a VAF of less than 1% in the same population, the effect size was increased (23% vs. 6%) [23]. The use of the lower cut-off VAFs allowed for the detection of a significant number of mutations that would have been missed in the studies mentioned above. Thirty-five of 86 (40%) of the PLWH and 10 of 32 (31%) of the non-HIV-infected individuals carried mutations with VAF between 0.5 and 1%, which may explain the identification of mutations less commonly associated with CH in this study, such as somatic *TP53* and *ARID1A*, which were only found in the PLWH [23]. Contrary to the studies by Dharan et al. and Bick et al., which identified *ASXL1* as one of the most commonly mutated genes in PLWH participants, Wang et al., did not detect mutations in this gene despite the lower VAF cut-off [23]. Moreover, the authors demonstrated that moderate-to-severe coronary artery stenosis was significantly more common in PLWH with CH than those without CH (30% vs. 9%) even after adjustment for the ACC-AHA Pooled Cohort Equation [23]. This finding is essential as it links CH and CVD among PLWH, supporting the hypothesis that CH promotes atherosclerotic disease in these individuals, likely through the induction of inflammatory alterations in the monocytes and migrated macrophages. The robustness of this study is limited, though, by the relatively small sample size, especially for the uninfected controls, and the significant differences in baseline demographics such as race, BMI, and lack of female participants.

Overall, there is robust epidemiological evidence that clonal hematopoiesis is more prevalent in PLWH than in non-HIV-infected individuals. Chronic inflammation can potentially mediate these associations. Numerous studies have highlighted that various inflammatory markers such as IL-6, soluble CD14, and CD163 are elevated among the PLWH compared to the uninfected individuals [27,28]. It is well described that sub-acute or chronic inflammation drives CH, likely by the increased resistance of mutated clones to inflammatory signals [29,30,31]. Thus, it is possible that induced inflammation in PLWH, potentially in combination with patient-related factors such as age and comorbidities, could increase the risk of CH development, further promoting inflammation and negatively affecting innate immunity [32] (Figure 1). Finally, the combined effect of CH, persistent chronic inflammation, and patient-related factors increase the risk of CVD [33,34,35] (Figure 1). 

However, the causality of these associations should be established with prospective studies adequately controlled for all possible confounders and adequately powered to detect a relationship between HIV infection duration, ART duration, viral load, or inflammatory markers. When designing such studies in the future, it is important to consider ways to decrease biases and maintain the generalizability of the results with broad participant eligibility criteria. Furthermore, technical details such as the sequencing starting material (whole blood vs. peripheral blood mononuclear cells), the sequencing ‘modalities’ sensitivity to detect low VAF, the variety of screened mutations, and the VAF cut-offs with the exclusion of germline mutations should be predetermined after weighing sensitivity with clinical meaningfulness of the findings. A large prospective study with long-term follow-up would be ideal to identify which aspect(s) of HIV infection drives this relationship with CH and the time course of the mutational landscape. Finally, this would allow the investigation of a possible impact of CH in developing CVD in PLWH as a mediator or inducer of inflammatory changes in the monocytes in these individuals, causing high-risk atherosclerotic lesions. 

## 3. Antiretroviral Therapy and CH

ART can be categorized into different major classes and the most commonly used include non-nucleoside reverse transcriptase inhibitors (NNRTIs), nucleoside reverse transcriptase inhibitors (NRTIs), protease inhibitors (PI) and integrase inhibitors (INSTI) [36]. The effect in hematopoiesis of some ART regimens has initially been explored in a preclinical murine model. The combinations of tenofovir disoproxil fumarate (TDF) plus lamivudine (3TC) and efavirenz (EFV) plus zidovudine (AZT) and 3TC, resulted in genotoxic and mitogenic effects in the bone marrow of mice that could have the potential for carcinogenesis [37]. However, no studies have explored those effects in humans. Focusing on AZT as an agent alone, an older NRTI used for the treatment of HIV, and the prevention of HIV transmission from actively viremic HIV+ mothers to their offspring during labor has been associated with myelotoxicity in humans [36]. A study by Shah et al. in 1996 showed that in vitro exposure of human hematopoietic progenitors in AZT could impact the erythroid progenitors but not the multipotent progenitors [38]. Indeed, AZT has been identified as a cause of pure red cell aplasia in a case report [39]. In a more recent study, Lin et al. investigated whether AZT could result in CH after in-utero exposure by examining 1-week-old neonate peripheral blood mononuclear cells for single nucleotide variants, small insertions and deletions, and large somatic copy number alterations [40]. The authors did not identify any significant difference in their mutational signature of AZT-exposed vs. AZT-unexposed infants [40].Therefore, since CH is linked to mutagenic events, the authors concluded that AZT exposure in-utero was not associated with CH and more studies on AZT and other ART regimens are imperative to safely determine their effect on CH [40]. 

More recently, Van der Heijden et al. found in his cross-sectional study that SBS18, a signature predominantly characterized by C>A mutations, was similarly identified as contributing to CH mutations in PLWH with prior exposure to AZT, whereas it was absent in unexposed individuals [24]. In the same study, a similar trend was noted for PIs which did not reach statistical significance [24]. A previous study published in the 2000s has shown that PIs may overcome hematopoiesis inhibition by inhibiting the caspase-dependent apoptotic pathway [41], which can theoretically increases the risk of CH. 

Additionally, a number of preclinical studies have shown that NRTIs can induce mitochondrial stress though the depletion of mitochondrial DNA [42,43,44,45]. It was recently highlighted that mitochondrial stress could negatively affect the function of hematopoietic stem cells and expedite their aging potentially via the upregulation of the inflammatory pathways [46,47]. Thus, it is possible that CH may develop as a protective mechanism against increased mitochondrial stress in the hematopoietic stem cells of individuals who are on NRTIs. However, as the epidemiologic evidence is lacking, further mechanistic studies are required to confirm this hypothesis. 

## 4. Effect of CH on the Course of HIV and Other Infections

While several recent studies have investigated the correlation between HIV infection and the development of CH [18,22,23,24], the associations of CH with the characteristics and the course of HIV infection and the risk for other infections in PLWH are less well known. 

Van der Heijden explored the clinical correlates of PLWH with their CH mutation carrier status and found that the CH carriers were older with a longer duration of HIV infection and lower CD4 nadir [24]. On the contrary, no significant difference in CH prevalence was found concerning CD4/CD8 T-cell ratio and the most recent CD4 T-cell count [24]. Older age, lower CD4 nadir, and increased CD4/CD8 ratio were independently associated with the CH mutation prevalence, while HIV duration and the most recent CD4 T cell count were not [24]. The same study demonstrated that the PLWH with a CH mutation had at least once a detectable HIV viral load within a year before a visit supporting an existing HIV reservoir [24]. Similarly, the ratio of HIV cell-associated (CA)-RNA to CA-DNA, which represents the relative viral transcription level, was found to be increased in the CH-mutation carriers compared to the PLWH without the CH mutation [24]. Overall, these observations support that CH can negatively affect the course of HIV infection in PLWH. 

Recent studies provided evidence that CH increases the risk of several other infections. Specifically, Bolton et al. found that CH increased the risk of *Streptococcal* and *Clostridium difficile* infections in patients with solid malignancies after adjusting for age, race, smoking, gender, cumulative exposure to cytotoxic therapy before the blood draw, cumulative exposure to cytotoxic therapy after the blood draw, and primary tumor site [48]. In the same study, the authors found that among COVID-19 positive patients, individuals with CH had a significantly higher risk for severe disease in a multivariable analysis showing a positive association between the VAF and the severity of illness [48]. The exact mechanisms underlying these associations remain unclear. Somatic alterations in the hematopoietic stem cells lead to altered inflammatory signaling in differentiated white blood cells, including the monocytes, migrated macrophages, and other cells participating in the innate immune reactions [49,50]. Thus, CH may affect the regulation of the innate immunity in the setting of infections leading to induced inflammatory responses with poor antimicrobial effect. Further preclinical studies and prospective clinical trials are required to shed light on the underlying biologic mechanisms linking CH to poor infection outcomes. 

## 5. The Role of CH in the Development of Myeloid Neoplasms in PLWH and the Outcomes of PLWH Who Develop Myeloid Neoplasms

Identifying the subset of the PLWH who eventually develop a myeloid malignancy and analyzing their mutational status is an indirect way to examine the role of CH in the development of hematopoietic malignancy in these patients. While some of the molecular abnormalities associated with CH seem to be enriched in the PLWH who develop MDS compared with the HIV-negative counterparts (mainly *ASXL1*, *DNMT3A* and *TP53* mutations, and higher risk cytogenetics) [26], progression of CH from >2% VAF and correlation with hematologic malignancy is not well defined [51]. The common mutations associated with CH (epigenetic factors *DNMT3A*, *ASXL1*, *TET2*, DNA damage repair genes such as *PPM1D*, *TP53*, signaling genes such as *JAK2*, and spliceosome components *SF3B1*, *SRSF2*) can act as drivers, more often in the PLWH who develop MDS than those who develop AML, where the most common alterations are adverse/intermediate karyotype or chromosome 7 abnormalities [52]. The role of other quantitative rather than qualitative factors, such as the role of the VAF percentage and the accumulation of more than one driver mutations, or the presence of CH without driver mutations in the PLWH who eventually develop a malignancy, should be taken into account when planning prospective follow-up studies of PLWH to understand the pathophysiology and need for surveillance of these patients. 

The outcomes of PLWH who develop myeloid neoplasms such as MDS and AML are worse than the non-HIV patients [26,53]. It has been reported that patients with HIV and MDS (HIV+/MDS) exhibited more prominent cytopenias and a higher percentage of marrow blasts, a marker indicative of a higher risk for AML transformation, than the HIV-/MDS patients [26,53]. Similarly, despite being younger and potentially better candidates for chemotherapy and allogeneic bone marrow transplantation, the HIV+/AML patients have worse overall survival outcomes compared to the HIV-/AML patients [54,55]. Since these findings suggest that the development of myeloid neoplasms among PLWH is associated with particularly poor outcomes and CH in its turn predisposes to myeloid neoplasms, the need for better understanding of the biology of these associations and the early detection of PLWH with CH to facilitate close monitoring and potentially early treatment to prevent the progression to advanced myeloid neoplasms is warranted. 

## 6. Conclusions

The optimization of ART has tremendously improved the outcomes of PLWH, who have now similar survival to the general population. However, PLWH have a higher prevalence of CVD and non-AIDS associated malignancies, with the underlying biology remaining unclear. The recent data support that PLWH have a higher prevalence of CH, with a potentially different pattern of acquisition of somatic mutations in their hematopoietic stem cells, which appears to be associated with a higher risk of CVD. The underlying biologic mechanism remains unclear, however HIV infection may increase the age-related replication stress to hematopoietic cells leading to a higher risk of acquisition of somatic mutations. Interestingly, CH among the PLWH is associated with lower CD4 nadir and worse HIV-related outcomes, potentially associated with worse infectious outcomes among individuals with CH. Finally, CH carries a significant risk of progression to myeloid neoplasms such as MDS and AML, which are strongly associated with worse outcomes among patients with HIV infection. Further basic science, translational studies, and prospective clinical trials are required to confirm these associations and elucidate the underlying biological mechanisms. 

## Figures and Tables

**Figure 1 cells-12-00686-f001:**
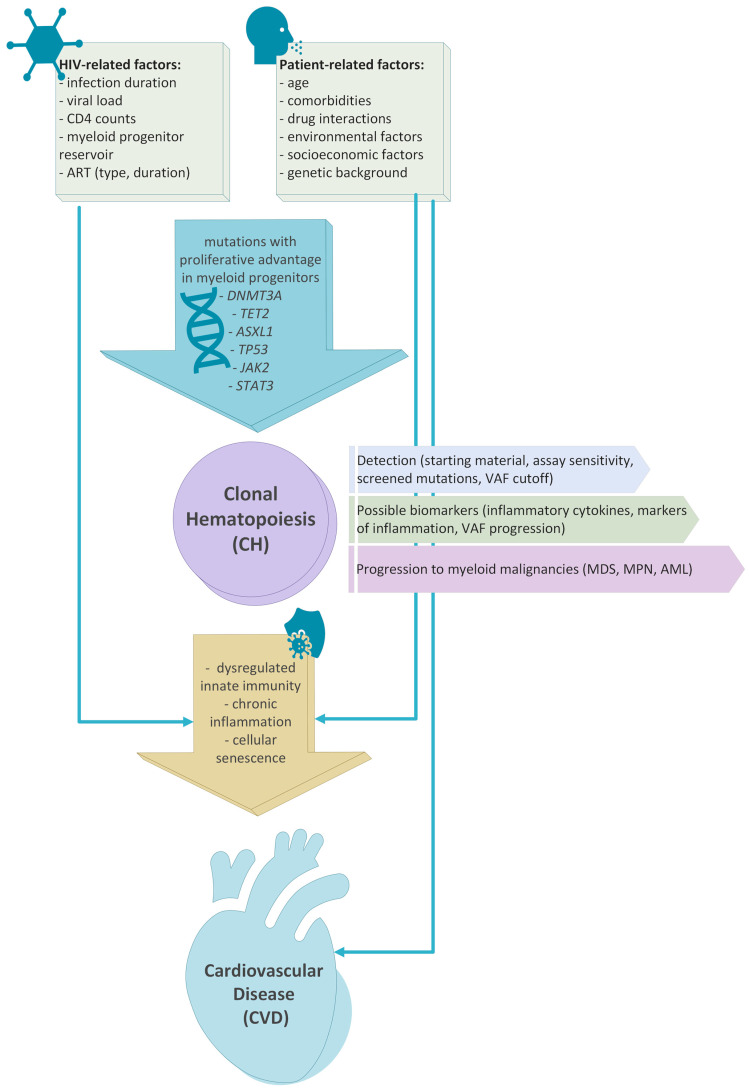
Association between HIV infection, patients’ characteristics, development of clonal hematopoiesis and the risk of development cardiovascular disease and myeloid neoplasms. HIV infection and patient-related risk factors such as age, comorbidities and medications have been associated with de-regulated innate immunity and chronic inflammation. These biologic processes have also been linked to clonal hematopoiesis. Clonal hematopoiesis and chronic inflammation are both associated with high risk of cardiovascular disease development. Finally, patients with clonal hematopoiesis have a higher risk of progression to myeloid neoplasms such as MDS, MPN and AML.

## Data Availability

Not applicable.

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
