# Peer review of "Human Immunodeficiency Virus and Clonal Hematopoiesis"

_cells, 2023, doi:10.3390/cells12050686_

Round 1
Reviewer 1 Report
Review of manuscript entitled Human Immunodeficiency Virus: clonal hematopoiesis and myeloid neoplasms.
Clonal hematopoiesis has been a hot-bed at the intersection of several diseases and several reviews have been published correlating it with cardiovascular disease, valvular dysfunction, diabetes, response to infections, neoplasms, especially hematopoietic, among others. It’s association with HIV has been the interest of several recent papers. The contents of this manuscript are interesting and, although reviewed previously, they are presented in a haphazard manner in the current manuscript. I shall highlight points that may need revision, correction, reformatting, or elaboration.
1. The manuscript title indicates that the contents are related to clonal hematopoiesis in “myeloid malignancies” in HIV patients. However, this forms only a subsection of the paper, discussed at the end
2. HIV has to be spelled out the first time (line 1, introduction)
3. PLWH has at many instances been mistyped as PWLH.
4. Under subheading “Clonal hematopoiesis is patients living with HIV,” line 9, covariate should perhaps read as correlation.
5. The conclusion of the ARCHIVE study quote in the subsequent sentence: “These findings suggest that PLWH have higher risk of developing CH,” should have some stronger evidence than correlational. The number of patients studied in this study (220 HIV vs 226 non-HIV) are small. Was any correlation made with prevalence of cardiovascular disease in this study?
6. The role of chronic inflammation in development of CH is rightly pointed out by the authors. This could be elaborated on more given this is likely the most plausible reason for a higher association of CH in PLWH.
7. Mitochondrial stress induced by ART has be conjectured as a possible cause of its association with CH in HIV patients. The authors may wish to add this information and mechanism in their paper.
8. Page 6, para 2, line 8: Perhaps the sentence Using the cut-off of 1%... should read as …less than 1%.
9. Page 7, last paragraph, regarding study by Van der Heijden the authors indicate that PLWH who were CH carriers were older with lower CD4 nadir. Subsequently, in the same paragraph (page 8) the text reads that CH mutation prevalence was independent of age and lower CD4 nadir.
10. How does the effect of CH on Covid severity and increased risk of other infections fit with the title of the paper? Perhaps the title needs to be broadened in its scope.
11. The authors need to expand on the role of CH in PLWH who develop myeloid neoplasms, for example, the role of specific common genes mutated in CH and whether they act as drivers, role of multiple mutations, higher VAF in development of myeloid neoplasms, and outcome. A good reference is: Acute Myeloid Leukemia in Patients Living with HIV Infection: Several Questions, Fewer Answers. nt. J. Mol. Sci. 2020, 21(3), 1081; https://doi.org/10.3390/ijms21031081
12. Page 10, under subheading MPN, line 9 should read as…be inferior “in” HIV+/CML…
Author Response
Reviewer 1
Clonal hematopoiesis has been a hot-bed at the intersection of several diseases and several reviews have been published correlating it with cardiovascular disease, valvular dysfunction, diabetes, response to infections, neoplasms, especially hematopoietic, among others. It’s association with HIV has been the interest of several recent papers. The contents of this manuscript are interesting and, although reviewed previously, they are presented in a haphazard manner in the current manuscript. I shall highlight points that may need revision, correction, reformatting, or elaboration.
- The manuscript title indicates that the contents are related to clonal hematopoiesis in “myeloid malignancies” in HIV patients. However, this forms only a subsection of the paper, discussed at the end
We thank the reviewer for bringing up this point. We performed an extensive revision of our review article and minimized the section describing the outcomes of myeloid neoplasms among patients with HIV. We describe the overall poor outcomes of MDS/AML patients with HIV to highlight the need for better understanding of the underlying biology and early detection of HIV patients who develop CH. We have changed our title to “Clonal hematopoiesis and HIV”.
- HIV has to be spelled out the first time (line 1, introduction)
We thank the reviewer for the comment. We performed extensive and more detailed proof reading and have incorporated the proposed changes.
- PLWH has at many instances been mistyped as PWLH.
We thank the reviewer for the comment. We performed extensive and more detailed proof reading and have incorporated the proposed changes.
- Under subheading “Clonal hematopoiesis is patients living with HIV,” line 9, covariate should perhaps read as correlation.
We agree with the reviewer. We have incorporated the proposed change.
- The conclusion of the ARCHIVE study quote in the subsequent sentence: “These findings suggest that PLWH have higher risk of developing CH,” should have some stronger evidence than correlational. The number of patients studied in this study (220 HIV vs 226 non-HIV) are small. Was any correlation made with prevalence of cardiovascular disease in this study?
We agree with the reviewer that evidence from the ARCHIVE study is observational/correlational. We have edited the manuscript to express the level of evidence supported from the study design (lines ). Regarding cardiovascular disease in this study population, the prevalence of cardiovascular comorbidities was similar between PLWH and controls (64.1 vs 65.5%). Authors evaluated the prevalence of a mutation in the IL-6 receptor (IL6R p.Asp358Ala), suggested by a different study to play a role in CH-related cardiovascular risk, but found no difference between the two groups. No other correlation or sub-analysis of CVD was made in this study. We incorporated the above findings as well (lines 83 – 90).
- The role of chronic inflammation in development of CH is rightly pointed out by the authors. This could be elaborated on more given this is likely the most plausible reason for a higher association of CH in PLWH.
We thank the reviewer for bringing up this important point. As a response we have included an additional paragraph and a figure (Figure 1) analyzing how chronic inflammation could mediate the association between HIV infection and development of CH and then mediate the development of CV disease. Lines 144 – 152 and 465 – 472.
- Mitochondrial stress induced by ART has be conjectured as a possible cause of its association with CH in HIV patients. The authors may wish to add this information and mechanism in their paper.
We thank the reviewer for this comment. As a response we included a section describing the findings that ART has been found to increase mitochondrial stress and that mitochondrial stress in hematopoietic cells is associated with expedited aging and dysfunction which increases the risk of CH development. Lines 190 – 196.
- Page 6, para 2, line 8: Perhaps the sentence Using the cut-off of 1%... should read as …less than 1%.
We thank the reviewer for the comment. We have incorporated the proposed change.
- Page 7, last paragraph, regarding study by Van der Heijden the authors indicate that PLWH who were CH carriers were older with lower CD4 nadir. Subsequently, in the same paragraph (page 8) the text reads that CH mutation prevalence was independent of age and lower CD4 nadir.
We thank the reviewer for bringing up this point. Based on the findings of Van der Heijden et al, age, lower CD4 nadir and increased CD4/CD8 ratio were independently associated with CH mutation prevalence in this cohort. We have revised our manuscript accordingly. Lines 205 – 207.
- How does the effect of CH on Covid severity and increased risk of other infections fit with the title of the paper? Perhaps the title needs to be broadened in its scope.
We agree with the reviewer’s point. We included this section to describe the effect of CH on the course of HIV and other infections highlighting that both CH and HIV infection have significant impact on immune dysregulation. As a response to this comment we have changed the title of this section to: “Effect of CH on the course of HIV and other infections”. Line 198.
- The authors need to expand on the role of CH in PLWH who develop myeloid neoplasms, for example, the role of specific common genes mutated in CH and whether they act as drivers, role of multiple mutations, higher VAF in development of myeloid neoplasms, and outcome. A good reference is: Acute Myeloid Leukemia in Patients Living with HIV Infection: Several Questions, Fewer Answers. nt. J. Mol. Sci. 2020, 21(3), 1081; https://doi.org/10.3390/ijms21031081
We agree with the reviewer that the role of CH in PLWH should be examined towards understanding of development of myeloid malignancies, All the factors mentioned in the reviewer’s comment should be incorporated in prospective studies that aim to delineate these relationships. We have included a section describing these findings. Lines 226 – 241.
- Page 10, under subheading MPN, line 9 should read as…be inferior “in” HIV+/CML…
We thank the reviewer for the comment. As a response to the comments from both reviewers we have revised our review extensively and minimized the section describing the outcomes of myeloid neoplasms among HIV patients. We describe the overall poor outcomes of MDS/AML patients with HIV to highlight the need for better understanding of the underlying biology and early detection of HIV patients who develop CH. Thus, we have removed the section describing the outcomes of HIV patients who develop MPN.

Reviewer 2 Report
In this Review, the authors summarized the current research on the association between HIV infection and clonal hematopoiesis, as well as the worse outcomes of patients with myeloid neoplasms. However, the manuscript was not written well, and the following points should be addressed prior to publication:
1. It would be helpful if the authors could draw a figure to illustrate the association of HIV infection with clonal hematopoiesis, inflammation status changes, elevating cardiovascular risk, and myeloid neoplasms.
2. The author implied that antiretroviral therapies might be involved in clonal hematopoiesis or the treatments on myeloid neoplasms. However, there is no discussion about the major antiretroviral therapies on HIV infection. The classes of antiretroviral therapies and at least their adverse effects related to CH need to be discussed.
3. The logic of this review was not expressed clearly. The leading paragraphs or sentences of each section need to be modified to reflect the theme. e.g., 1) The first paragraph of “Effect of CH on the course of HIV infection” on page 7: the authors seemed to discuss the associations of CH with HIV infection, risk factors of CVD and cancer in PWLH, but the risk of other infections was discussed in this section stead, which is confusing. 2) “MDS”, “MPN”, and “AML” are all subtitles of the section on page 8- “The effect of HIV infection on the outcomes of patients with myeloid neoplasms”. The full names should be spelled out as subtitles and a distinguishable format should be used for subtitles.
4. The manuscript was not written well and there are many editing errors and typos. e.g., Page 2, line 3: loving -> living; Page 2, line 14, literature -> literatures; Page 5, paragraph 3, line 2: 200HIV-> 200 HIV; Page 10: molecula r level-> molecular level…
Author Response
Reviewer 2
In this Review, the authors summarized the current research on the association between HIV infection and clonal hematopoiesis, as well as the worse outcomes of patients with myeloid neoplasms. However, the manuscript was not written well, and the following points should be addressed prior to publication:
- It would be helpful if the authors could draw a figure to illustrate the association of HIV infection with clonal hematopoiesis, inflammation status changes, elevating cardiovascular risk, and myeloid neoplasms.
We thank the reviewer for this suggestion, and we have inserted an illustrated flowchart summarizing the association of HIV infection with clonal hematopoiesis, inflammation and risk of developing cardiovascular disease and myeloid neoplasms. Lines 144 – 152 and 465 – 472.
- The author implied that antiretroviral therapies might be involved in clonal hematopoiesis or the treatments on myeloid neoplasms. However, there is no discussion about the major antiretroviral therapies on HIV infection. The classes of antiretroviral therapies and at least their adverse effects related to CH need to be discussed.
We appreciate the reviewer’s comment, as we think this would be a great and essential addition to our review. We mention the classes of the ART and indicate the studies that have explored the role of zidovudine as myelotoxic or potentiator of CH. To-date, other medications/classes of ART have not yet been identified or investigated for any implication in CH. Lines 165 – 196.
- The logic of this review was not expressed clearly. The leading paragraphs or sentences of each section need to be modified to reflect the theme. e.g.,
1) The first paragraph of “Effect of CH on the course of HIV infection” on page 7: the authors seemed to discuss the associations of CH with HIV infection, risk factors of CVD and cancer in PWLH, but the risk of other infections was discussed in this section stead, which is confusing.
We thank the reviewer for the comment. We have opted to include this small section discussing susceptibility to other infections to point out the possible implications of CH on immune dysregulation and chronic inflammation, affecting not only the outcomes of patients with HIV but other infections including COVID19 based on emerging data. As a response we have changed the title of this section to: “Effect of CH on the course of HIV and other infections”. Line 198.
2) “MDS”, “MPN”, and “AML” are all subtitles of the section on page 8- “The effect of HIV infection on the outcomes of patients with myeloid neoplasms”. The full names should be spelled out as subtitles and a distinguishable format should be used for subtitles.
We thank the reviewer for the comment. We have revised our review extensively to focus on CH and HIV and thus minimized the section describing the outcomes of myeloid neoplasms among patients with HIV. We describe the overall poor outcomes of MDS/AML patients with HIV to highlight the need for better understanding of the underlying biology and early detection of HIV patients who develop CH. Thus, we have removed the MDS/MPN/AML subtitles.
- The manuscript was not written well and there are many editing errors and typos. e.g., Page 2, line 3: loving -> living; Page 2, line 14, literature -> literatures; Page 5, paragraph 3, line 2: 200HIV-> 200 HIV; Page 10: molecula r level-> molecular level…
We thank the reviewer for the comment. We performed extensive and more detailed proof reading and have corrected any existent grammatical errors.

Round 2
Reviewer 2 Report
My previous comments have been properly addressed. In my opinion, the revised manuscript can be published in Cells.